

# Comparative analysis of the liver transcriptome in the red-eared slider *Trachemys scripta elegans* under chronic salinity stress

Meiling Hong[1], Aiping Jiang[1], Na Li[1], Weihao Li[1], Haitao Shi[1,2], Kenneth B. Storey[3] and Li Ding[1]

[1] Ministry of Education Key Laboratory for Ecology of Tropical Islands, College of Life Sciences, Hainan Normal University, Haikou, Hainan, China
[2] Chengdu Institute of Biology, Chinese Academy of Sciences, Chengdu, China
[3] Department of Biology, Carleton University, Ottawa, Canada

## ABSTRACT

The red-eared slider (*Trachemys scripta elegans*), identified as one of the 100 most invasive species in the world, is a freshwater turtle originally from the eastern United States and northeastern Mexico. Field investigations have shown that *T. s. elegans* can survive and lay eggs in saline habitats. In order to understand the molecular mechanisms of salinity adaptation, high-throughput RNA-Seq was utilized to identify the changes in gene expression profiles in the liver of *T. s. elegans* in response to elevated salinity. We exposed individuals to 0, 5, or 15 psu (practical salinity units) for 30 days. A total of 157.21 million reads were obtained and assembled into 205138 unigenes with an average length of 620 bp and N50 of 964 bp. Of these, 1019 DEGs (differentially expressed genes) were found in the comparison of 0 vs. 5 psu, 1194 DEGs in 0 vs. 15 psu and 1180 DEGs in 5 vs. 15 psu, which are mainly related to macromolecule metabolic process, ion transport, oxidoreductase activity and generation of precursor metabolites and energy by GO (Gene Ontology) enrichment analyses. *T. s. elegans* can adapt itself into salinity by balancing the entry of sodium and chloride ions via the up-regulation expression genes of ion transport (potassium voltage-gated channel subfamily H member 5, *KCNH5*; erine/threonine-protein kinase 32, *STK32*; salt-inducible kinase 1, *SIK1*; adiponectin, *ACDC*), and by accumulating plasma urea and free amino acid via the up-regulation expression genes of amino acid metabolism (ornithine decarboxylase antizyme 3, *OAZ3*; glutamine synthetase, *GLUL*; asparaginase-like protein 1b, *ASRGL*; L-amino-acid oxidase-like, *LAAO*; sodium-dependent neutral amino acid transporter B, *SLC6A15s*; amino acid permease, *SLC7A9*) in response to osmotic regulation. An investment of energy to maintain their homeostatic balance is required to salinity adaptation, therefore, the genes related to energy production and conversion (F-ATPase protein 6, *ATP6*; cytochrome c oxidase subunit I, *COX1*; cytochrome c oxidase subunit III, *COX3*; cytochrome b, *CYTb*; cytochrome P450 17A1, *CYP17A1*) were up-regulated with the increase of gene expression associated with lipid metabolism (apolipoprotein E precursor, *APoE*; coenzyme Q-binding protein, *CoQ10*; high-density lipoprotein particle, *SAA*) and carbohydrate metabolism (*HK*, *MIP*). These findings improve our understanding of the underlying molecular mechanisms involved in salinity adaptation

Corresponding author
Li Ding, dingli@hainnu.edu.cn

and provide general guidance to illuminate the invasion potential of *T. s. elegans* into saline environments.

## INTRODUCTION

The red-eared slider (*Trachemys scripta elegans*) has been introduced into diverse aquatic habitats worldwide (including many countries in Africa, Asia and Europe as well as Australia) via the pet-release pathway and, as a result, is classified as a highly invasive species by the International Union for Conservation of Nature (*Luiselli et al., 1997*; *Martins, Assalim & Molina, 2014*). It is native to freshwater habitats in 19 states of the eastern United States and two states of northeastern Mexico (*Mittermeier et al., 2015*). Recently these turtles have been found to lay eggs in the low salinity (0.1–26‰) estuary of the Nandujiang in Hainan Province, China (*Liu et al., 2011*; *Yang & Shi, 2014*). The extent of the saltwater adaptability of *T. s. elegans* is not fully understood. Studies of endocrine stress responses by *T. s. elegans* in the Lake Pontchartrain Basin of Louisiana suggest that these turtles may serve as a sentinel species for elevated salinity in environments where salinity is rising due to saltwater intrusion (*Thompson, Franck & Valverde, 2011*). These studies indicate that *T. s. elegans* can invade not only fresh water, but also saline water environments. Therefore, the invasion potential and the mechanism of response to salinity adaptation of *T. s. elegans* is of serious concern.

Changing levels of salinity are a crucial environmental stress factor for aquatic species that can disrupt electrolyte balance, cell energetics, and various other physiological responses, including activating stress hormones (*Lushchak, 2011*). Species show altered composition in osmolality of body fluids in response to changing salinity (*Charmantier et al., 2011*) and adaptation to salinity change typically involves the physiological solution: tolerance of elevated inorganic ion concentrations (mainly sodium and chloride) in plasma (*Gordon & Tucker, 1965*), and accumulation of organic osmolytes (e.g., urea) to counteract cell-volume changes. The most dramatic changes in urea concentration are seen in plasma and tissues such as skeletal muscle, resulting from the up-regulation of hepatic urea (*Wright et al., 2004*). Accumulation of intracellular free amino acids via hepatic protein degradation or de novo amino acid synthesis can also contribute to osmotic balance (*McNamara et al., 2004*; *Yancey, 1985*). When subjected to ambient salinity change, *T. s. elegans* increased serum glucose levels, and the activities of creatine kinase (CK), aspartate aminotransferase (AST), lactate dehydrogenase (LDH), and alkaline phosphatase (ALP) in liver (*Shu et al., 2012*). Our previous studies have shown that *T. s. elegans* can increase blood osmotic pressure by balancing the entry of sodium and chloride ions with a decrease in the secretion of aldosterone, and by accumulating plasma urea for osmoregulation when ambient salinity was lower than 15‰ (*Hong et al., 2014*). However the molecular basis of these adaptive responses has not been studied in *T. s. elegans*.

Recently, high-throughput next-generation sequencing techniques have allowed researchers to broadly explore the extent and complexity of the transcriptomes of a wide range of eukaryotic species and to gain novel information about the gene responses of aquatic species to physiological and environmental stresses (*Wang et al., 2009*). RNA-seq is an efficient technique to probe the gene responses to physiological stress (*Li et al., 2013*; *Smith, Bernatchez & Beheregaray, 2013*; *Xia et al., 2013*), particularly when working with species that do not have a sequenced genome. The present study used RNA-seq to analyze the transcriptomic response of *T. s. elegans* to salinity stress and identify the genes (and their metabolic functions) that are involved in salinity adaptation of turtles challenged by a brackish water environment. These results provide insights into the molecular mechanisms underlying osmoregulation in *T. s. elegans* and address the potential for this species to further invade and spread through new aquatic and brackish territories.

## MATERIALS AND METHODS

### Animals
Healthy *T. s. elegans* were obtained from a local turtle farm in Hainan Province, China and were acclimated in three cement pools half-filled with freshwater for two weeks. After acclimatization, nine healthy *T. s. elegans* (BW: 424–478 g, 2 years old) were divided into three groups in pools (190 cm × 65 cm × 32 cm) of differing salinity: one in freshwater serving as the control (0 practical salinity units, psu), and the other two challenged with 5‰ (5 psu) or 15‰ (15 psu) saltwater. Turtles were fed a commercial diet each Monday and Thursday and 24 h after feeding, unused feed was siphoned out followed by replacement of one-third of the water in each pool. Water salinity was measured every day and adjusted to the proper salinity as needed. Other water quality parameters were monitored 2–3 times a week with steady values of pH 7.5–7.9, total ammonia nitrogen of $<0.02$ mg $L^{-1}$ and temperature 26–28 °C. Photoperiod was 12 h:12 h L:D throughout.

After 30 days of exposure to the three experimental conditions, the three turtles from each group were subjected to a 24 h fast, then anesthetized by cryo-anesthesia moving turtles to −20 °C for 0.5–1 h. Experimental animal procedures had the prior approval of the Animal Research Ethics Committee of Hainan Provincial Education Centre for Ecology and Environment, Hainan Normal University (permit no. HNECEE-2014-004). Following euthanasia, the liver of each individual was sampled and divided into two sections, flash frozen in liquid nitrogen, and stored at −80 °C until used for RNA extraction. One liver section was used for quantitative real-time PCR (qRT- PCR), and the other was mixed from each group of three turtles for RNA-seq analysis.

### Total RNA extraction, library construction and sequencing
Extraction of total RNA from liver samples used TRIzol® Reagent, following manufacturer's instructions. Total RNA purity and concentration were determined using a NanoDrop 2000. The sequencing library was then constructed from high-quality RNA ($OD_{260/280} = 1.8$–2.2, $OD_{260/230} \geq 1.5$, RIN $\geq 8.0$, 28S:18 $\geq 1.0$, $>10$ μg). RNA-seq analysis was provided by Novel Bioinformatics Co., Ltd using the Sanger/Illumina method. Subsequently, cDNA libraries were made using the Hiseq4000 Truseq SBS Kit v3-HS

using 5 μg total RNA, following manufacturer's instructions. Poly(A) mRNA was isolated with Dyabeads (Life Technologies, USA), fragmented with RNaseIII and purified. The fragmented RNA was added and ligated with ion adaptor. Then double-stranded cDNA was synthesized and purified using magnetic beads. The molar concentration of the purified cDNA in each cDNA library was then quantified with a TBS-380 fluorometer using Picogreen. The paired-end RNA-seq library was sequenced with an Illumina HiSeq 4000. The RNA-Seq data were deposited in the NCBI with accession number GSE117354.

## De novo assembly and annotation

Raw reads were trimmed and quality controlled using SeqPrep (https://github.com/jstjohn/SeqPrep) and Sickle (https://github.com/najoshi/sickle) using default parameters. High-quality trimmed sequences were used for sequence assembly with Trinity (https://github.com/trinityrnaseq/trinityrnaseq/wiki) (Grabherr et al., 2011). Q20, Q30, GC-content and sequence duplication level of the clean data were all calculated. After that, All assembled transcripts were identified by using BLASTX against the databases of NR (NCBI non-redundant protein sequence), Swissprot (a manually annotated and reviewed protein sequence database), Pfam (Protein family), GO (Gene ontology), COG (Clutsters of Orthologous Groups of proteins) and KEGG (Kyoto Encyclopaedia of Genes and Genomes). The BLAST2GO (http://www.blast2go.com/b2ghome) program (Conesa et al., 2005) was used to obtain gene ontology (GO) annotations of unique assembled transcripts for describing biological processes, cellular components, and molecular functions.

## Analysis of differential expression and functional enrichment

Expression levels of transcripts were calculated as fragments per kilobase of exon per million mapped reads (FPKM). RNA-Seq by Expectation-Maximization (RSEM; http://deweylab.biostat.wisc.edu/rsem/) was used to quantify gene transcripts, and DEGseq (http://www.bioconductor.org/packages/release/bioc/vignettes/DEGseq/inst/doc/DEGseq.pdf) was used to conduct differential expression analysis. The resulting $p$ values were adjusted using the Benjamini and Hochberg's approach for controlling the false discovery rate. Genes with an adjusted $p$ value <0.05 found by DEGseq were assigned as differentially expressed. GO enrichment analysis of the DEGs was implemented by the GOseqR packages based on Wallenius non-central hyper-geometric distribution (Xie et al., 2011), which can adjust for gene length bias in DEGs.

## Experimental validation by qRT-PCR

Eighteen genes identified as significantly expressed from the GO terms related to osmotic regulation were selected to understand the gene expression levels in different groups, and also used for validation by qRT-PCR. Table 1 shows the specific primers used. TRIzol® Reagent (Invitrogen, Carlsbad, USA) was used to extract total RNA from liver, followed by reverse-transcription using First-strand cDNA Synthesis Kit (Invitrogen, Carlsbad, CA, US). The qRT-PCR for gene expression was analyzed by an Applied Biosystems 7500 Fast Real-Time PCR System in 96-well plates with a 20 μl reaction volume containing 1× SYBR Green qPCR Master Mix, gene-specific forward and reverse primers (0.4 μM) and cDNA (8 ng). The cycling conditions were 95 °C for 2 min followed by 40 cycles of 95 °C for 5 s

**Table 1  Sequences of primers for qRT-PCR validation.**

| Gene | Forward primer (5′ to 3′) | Reserse primer (5′ to 3′) | Product length (bp) |
|---|---|---|---|
| COX3 | TCACTTGAGCCCACCATAGC | AGAGCCGTACACACCATCAG | 155 |
| ATP6 | CTTGATGCCCTCTTCCCGTG | TTCCTCGTTCTCCACAGCCT | 145 |
| CYP17A1 | GATCGGCTTCGAGAGACACC | GGATCAGCAGAGGGGAGACA | 110 |
| APoE | GTCTGGAGCGGGCTTAGTAG | CATTCCCAGGTCTCCCACAG | 117 |
| CoQ10 | GCGAGTGCTGGGCTACT | TGAGCCCTTTGCGGTAGG | 119 |
| SAA | TCTAGGCGCTGGGGATATGT | CCACTAATGCCATCCTGCCA | 180 |
| FADS6 | GCTGCCATACAACGAGGACT | AGCCCTATGTCTTGCTGTCG | 146 |
| HK | TAAAGGCGTAACCAGGCTGC | AATCGCACGTCAGAGTCAGG | 126 |
| GCK | CGGGAACTGCTGAAATGCTC | GAATGTGAAGCCCAGAGGCA | 104 |
| GLUL | GTTGCCACACCAACTTCAGC | AAGCGCGGATATGGTACTGG | 111 |
| ASRGL | TAGCACCTGTTCCAGTGAGC | GCTGTGTTTGATGCAGGTCA | 193 |
| TAT | CATCCACCAGCGACTCCAAG | CATCCTGGTGCCAAGACCTG | 137 |
| ASS1 | CGGGCTGTACCAGAAACCAT | GGGACCATCCTGTACCATGC | 129 |
| STK32 | TCCAGTGCTAATGCCAGCTC | TGGAACACCCCTTCCTGGTT | 172 |
| SIK1 | TGGTGTGGTGCTGTATGTCC | TCCACAACTAGCATCCGTCG | 156 |
| SIK2 | GCTGGTCCTAGACCCATCCA | GAAGGCTCGTTCTCCTGTCC | 123 |
| INSRR | CCGAGTACCGTGATCTGCTC | GGCAGCTCCACATCTACCAC | 127 |
| STK33 | AGGCAGTTTTGGGGTGGTAA | TAAGATGCTCACCTCCCGTTC | 130 |
| β-actin | GCACCCTGTGCTGCTTACA | CACAGTGTGGGTGACACCAT | 190 |

and 60 °C for 30 s. *β-actin* was chosen as the reference gene, as it was expressed in the three groups and unaffected under salinity stress, and relative fold changes were determined using Relative Expression Software Tool v.2009 based on the cycle threshold (Ct) values generated by qRT-PCR.

The mRNA expression levels were expressed as mean ± standard error. Statistical analyses were conducted with SPSS 19.0. After testing the homogeneity of variance, statistical difference between treatments and controls were determined by one-way analysis of variance (ANOVA). LSD multiple comparison tests were carried out when the variances were homogeneous. Significant differences were set at $p < 0.05$.

# RESULTS

## Analysis of sequenced data quality

Because we haven't got the genome of *T. s. elegans* yet, a transcriptome was used as a reference to identify the differentially expressed genes induced by salinity exposure. Therefore, a mixed RNA pool from the samples of the control and salinity treatment groups was sequenced as the reference transcriptome. The data qualities from each sample were shown in Table 2. A total of 157.21 million reads and 23.58 billion bases were obtained from the liver transcriptome of *T. s. elegans*, including 50.68 million in 0 psu, 57.91 million in 5 psu group and 48.62 million in 15 psu group. After filtering low quality sequences by, trimming sequencing adapters/poly-N and removing poor quality reads, there were 152.52 million clean reads (97.02% of raw reads) were retained including 48.92 million in 0 psu,

**Table 2  Summary of Illumina expressed short reads production and filtering.**

|  | Groups | Total_Reads | Total_Bases | Error% | Q20% | Q30% | GC% |
|---|---|---|---|---|---|---|---|
| Raw data | 0 psu | 50681398 | 7602209700 | 0.0126 | 96.31 | 92.31 | 49.34 |
|  | 5 psu | 57912616 | 8686892400 | 0.0121 | 96.72 | 92.87 | 49.18 |
|  | 15 psu | 48616356 | 7292453400 | 0.0129 | 96.21 | 91.98 | 49.93 |
| Clean data | 0 psu | 48918412 | 7157032240 | 0.0104 | 98.29 | 95.12 | 49.19 |
|  | 5 psu | 56498176 | 8281257833 | 0.0103 | 98.34 | 95.18 | 49.09 |
|  | 15 psu | 47101028 | 6881004183 | 0.0107 | 98.17 | 94.77 | 49.83 |

**Table 3  Summary of *de novo* assembly results of Illumina sequence data.**

| Type | Unigene | Transcripts |
|---|---|---|
| Total sequence number | 205,138 | 244,815 |
| Total sequence base | 127,206,404 | 185,942,194 |
| Percent GC | 46.09 | 46.65 |
| Largest | 22,866 | 22,866 |
| Smallest | 201 | 201 |
| Average | 620.1 | 759.52 |
| N50 | 964 | 1,535 |
| N90 | 249 | 268 |

**Table 4  Functional annotation of the *Trachemys scripta elegans* transcriptome.**

| Annotated database | Number of unigenes | Precent(%) |
|---|---|---|
| Pfam | 20,458 | 9.97 |
| KEGG | 20,421 | 9.95 |
| GO | 13,362 | 6.51 |
| COG | 10,673 | 5.2 |
| Swissprot | 25,438 | 12.36 |
| NR | 38,651 | 18.84 |
| Total | 205,138 | 1 |

56.50 million in 5 psu and 47.10 million in 15 psu. Subsequently, 205,138 unigenes with an average length of 620 bp and N50 of 964 bp were obtained by de novo assembly. The largest and smallest unigenes were 22,866 bp and 201 bp, respectively (Table 3).

## Annotation and differential expression of genes

Putative functions of proteins encoded by the 205,138 genes were predicted by NR, Pfam, COG, Swissprot, GO and KEGG database. The results showed that there were 20,458 (9.97%), 20,421 (9.95%), 13,362 (6.51%), 10,673 (5.2%), 25,438 (12.36%), and 38,651 (18.84%) in Pfam, KEGG, GO, COG, Swissprot and NR databases, respectively (Table 4).

By GO annotation, the genes up-regulated or down-regulated were divided into three categories including biological process, cellular component and molecular function (Fig. 1). Among the category of biological process, the number of differentially expressed gene was higher in the GO terms of cellular process, metabolic process, single-organism process
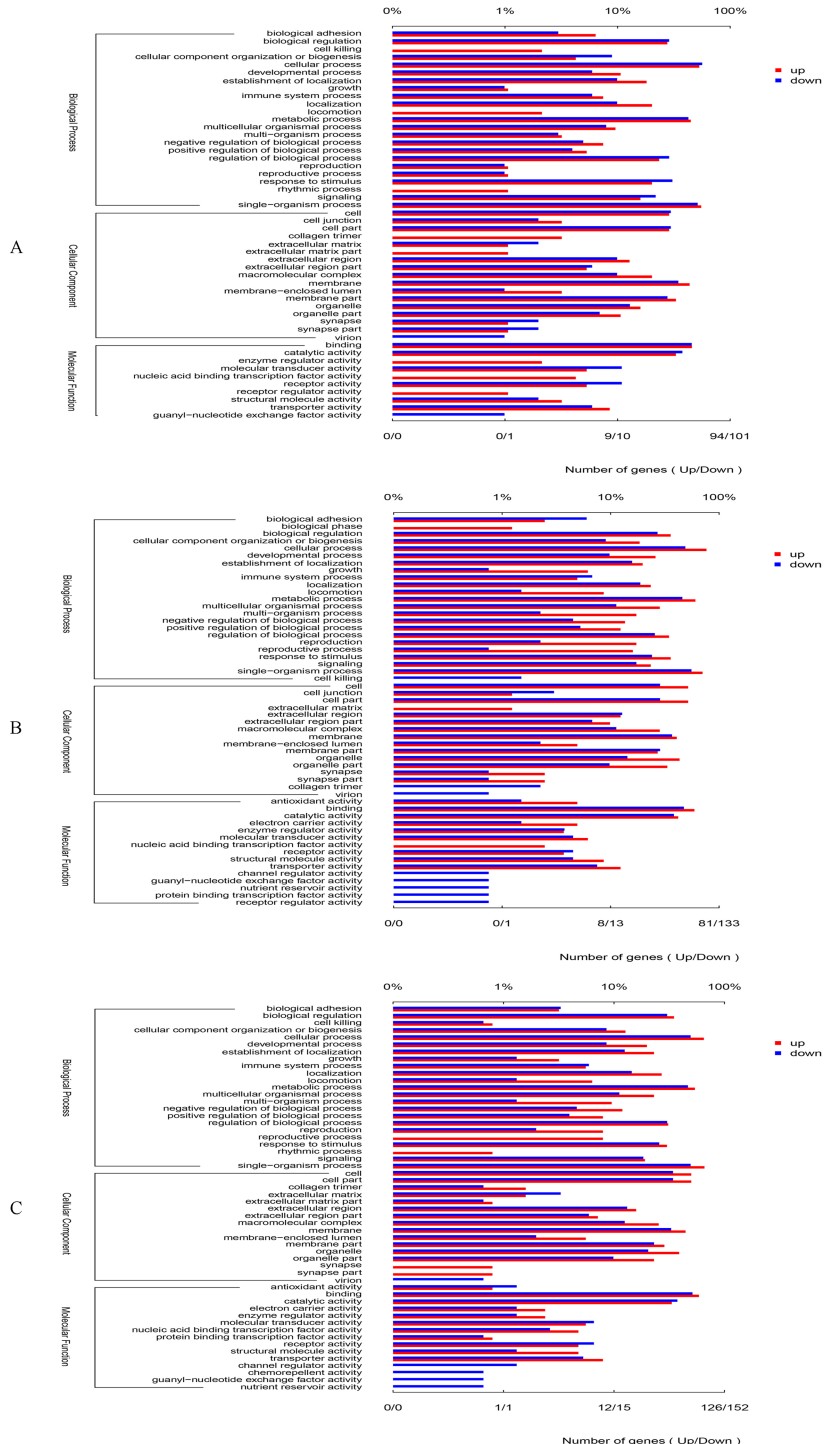

**Figure 1** **GO categories in the comparison of 0 vs. 5 psu (A), 0 vs. 15 psu (B), and 5 vs. 15 psu (C).** All genes were divided into several functional groups within three categories: cellular component, molecular function, and biological process. The below *x*-axis indicated the number of genes in each category, while the above *x*-axis indicated the percentage of total genes in that category.

and biological regulation. Among the category of cellular component, the number of differentially expressed gene was higher in the GO terms of cell, cell part, macromolecular complex, and membrane. In relation to molecular function, the number of differentially expressed gene was higher in the GO terms of binding, catalytic activity, and transporter activity. Moreover, there was almost the same trend in the comparisons of 0 vs. 5 psu, 0 vs. 15 psu, and 5 vs. 15 psu.

On the basis of criteria of two-fold or greater change and Q of $p < 0.05$, 3,393 unigenes were identified as significant differentially expressed genes (DEGs), including 1019 (445 up-regulated and 574 down-regulated) DEGs in 0 vs. 5 psu, 1,194 (526 up-regulated and 668 down-regulated) DEGs in 0 vs. 15 psu, and1180 (548 up-regulated and 632 down-regulated) DEGs in 5 vs. 15 psu. Many DEGs were classified into some dominant categories by GO enrichment analyses, including the macromolecule metabolic process (52 DEGs), ion transport (35 DEGs), ion transmembrane transport (24 DEGs) in biological process, and nucleic acid binding (37 DEGs), oxidoreductase activity (35 DEGs), transporter activity (33 DEGs), and transmembrane transporter activity (27 DEGs) in molecular function (Table 5).

As for osmoregulation, there were 15 major GO terms related to osmotic function in turtle liver in the comparison of 0 vs. 5 psu (Table S1) and 12 in the comparison of 0 vs. 15 psu (Table S2). Based on GO enrichment analysis, a list of candidate genes involved in salinity acclimation in *T. s. elegans* were identified. Many of these were identified as encoding proteins involved in ion transport, energy production and conversion, and macromolecule metabolic process including lipid, protein and carbohydrate.

Based on the annotation of DEGs, we selected ten genes related to ion regulation, five genes related to energy production and conversion, five genes related lipid metabolism, eight genes related amino acid metabolism and six genes related carbohydrate metabolism (Table 6). Among the DEGs associated with ion regulation, five genes (adipocytokine, *ACDC*; insulin receptor-related protein, *INSRR*; serine/threonine-protein kinase 32, *STK32*; salt-inducible kinase 1, *SIK1*; potassium voltage-gated channel subfamily H member 5, *KCNH5*) were up-regulated in the comparison of 0 vs. 5 psu, and four genes (salt-inducible kinase 2, *SIK2*; *SIK1*; *ACDC*; *STK32*) were up-regulated in the comparison of 0 vs. 15 psu, and five genes (*SIK1*, *SIK2*, *STK33*, *ACDC*, and solute carrier family 26 member 9, *SLC26A9*) were up-regulated in the comparison of 5 vs. 15 psu. All of the five DEGs (cytochrome c oxidase subunit I, *COX1*; cytochrome c oxidase subunit III,*COX3*; cytochrome b, *CYTb*; F-ATPase protein 6, *ATP6*, and cytochrome P450 17A1, *CYP17A1*) associated with energy production and conversion showed up-regulation under salinity stress, which indicated that there was a higher energy demand in response to salinity exposure. In general, lipid and carbohydrate were the main sources of energy. As for five DEGs associated with lipid metabolism, three DEGs (apolipoprotein E precursor, *ApoE*; coenzyme Q-binding protein, *CoQ10*; high-density lipoprotein particle, *SAA*) increased with ambient salinity increased, while one DEGs (alcohol dehydrogenase 4, *ADH4*) showed down-regulation in the comparison of 0 vs. 5 psu and one DEGs (fatty acid desaturase 6, *FADS6*) down-regulation in the comparison of 0 vs. 15 psu. Among six DEGs related to carbohydrate metabolism, two DEGs (hexokinase, *HK* and lens

**Table 5  Summary of GO term enrichment results on ion-regulation and macromolecular metabolism in *T. s. elegans* under salinity stress.**

| GO ID | Description | Ratio in study | Ratio in pop | *p*-value (FDR) |
|---|---|---|---|---|
| GO:0006091 | generation of precursor metabolites and energy | 17/357 | 119/13,362 | 4.33E − 05 |
| GO:0009060 | aerobic respiration | 7/357 | 17/13,362 | 0.0003 |
| GO:0045333 | cellular respiration | 7/357 | 19/13,362 | 0.0004 |
| GO:0015980 | energy derivation by oxidation of organic compounds | 8/357 | 40/13,362 | 0.0065 |
| GO:0034220 | ion transmembrane transport | 24/357 | 336/13,362 | 0.0081 |
| GO:0098660 | inorganic ion transmembrane transport | 18/357 | 209/13,362 | 0.0081 |
| GO:0098655 | cation transmembrane transport | 18/357 | 210/13,362 | 0.0081 |
| GO:0006119 | oxidative phosphorylation | 4/357 | 7/13,362 | 0.0081 |
| GO:1902991 | regulation of amyloid precursor protein catabolic process | 3/357 | 3/13,362 | 0.0081 |
| GO:1902992 | negative regulation of amyloid precursor protein catabolic process | 3/357 | 3/13,362 | 0.0081 |
| GO:1902430 | negative regulation of beta-amyloid formation | 3/357 | 3/13,362 | 0.0081 |
| GO:1902003 | regulation of beta-amyloid formation | 3/357 | 3/13,362 | 0.0081 |
| GO:0006811 | ion transport | 35/357 | 623/13,362 | 0.0109 |
| GO:0006812 | cation transport | 23/357 | 335/13,362 | 0.0114 |
| GO:1902600 | hydrogen ion transmembrane transport | 10/357 | 77/13,362 | 0.0117 |
| GO:1900221 | regulation of beta-amyloid clearance | 3/357 | 4/13,362 | 0.0177 |
| GO:0098662 | inorganic cation transmembrane transport | 15/357 | 176/13,362 | 0.0178 |
| GO:0050773 | regulation of dendrite development | 6/357 | 28/13,362 | 0.0184 |
| GO:0015672 | monovalent inorganic cation transport | 14/357 | 171/13,362 | 0.0410 |
| GO:0005576 | extracellular region | 18/357 | 220/13,362 | 0.0097 |
| GO:0044463 | cell projection part | 16/357 | 206/13,362 | 0.0281 |
| GO:0015078 | hydrogen ion transmembrane transporter activity | 11/357 | 76/13,362 | 0.0047 |
| GO:0015077 | monovalent inorganic cation transmembrane transporter activity | 16/357 | 176/13,362 | 0.0083 |
| GO:0022890 | inorganic cation transmembrane transporter activity | 18/357 | 236/13,362 | 0.0167 |
| GO:0008324 | cation transmembrane transporter activity | 20/357 | 293/13,362 | 0.0268 |
| GO:1902991 | regulation of amyloid precursor protein catabolic process | 3/357 | 3/13,362 | 0.0081 |
| GO:1902992 | negative regulation of amyloid precursor protein catabolic process | 3/357 | 3/13,362 | 0.0081 |
| GO:0034364 | high-density lipoprotein particle | 5/357 | 13/13,362 | 0.0081 |
| GO:0032994 | protein-lipid complex | 5/357 | 16/13,362 | 0.0129 |
| GO:1990777 | lipoprotein particle | 5/357 | 16/13,362 | 0.0129 |
| GO:0034358 | plasma lipoprotein particle | 5/357 | 16/13,362 | 0.0129 |

**Notes.**

GO names were retained only from GO terms of levels >2.

fiber major intrinsic protein, *MIP*) increased with ambient salinity increased, four DEGs (glucokinase, *GCK*; tagatose 1,6-diphosphate aldolase, *LacD*; L-gulono-gamma-lactone oxidase, *GLO* and ribulose bisphosphate carboxylase small chain, *RBCs*) decreased under salinity stress. In addition, six DEGs (ornithine decarboxylase antizyme 3, *OAZ3*; glutamine synthetase, *GLUL*; asparaginase-like protein, *ASRGL*; L-amino-acid oxidase-like, *LAAO*; sodium-dependent neutral amino acid transporter, *SLC6A15s*; amino acid permease, *SLC7A9*) related to amino acid metabolism showed up-regulation and two DEGs (tyrosine

**Table 6  DEGs related to energy production and conversion, macromolecule metabolic process, and ion transport in the liver of *T. s. elegans*.**

| Unigene ID | Description | Log$_2$ (5/0 psu) | Log$_2$ (15/0 psu) | Log$_2$ (15/5 psu) |
|---|---|---|---|---|
| **DEGs related to energy production and conversion** | | | | |
| c198757_g1 | cytochrome c oxidase subunit I (*COX1*) | 7.03 | 7.14 | – |
| c205807_g1 | cytochrome c oxidase subunit III (*COX3*) | 5.79 | 5.57 | – |
| c184332_g1 | cytochrome b (*CYTb*) | 6.37 | 6.51 | – |
| c168776_g1 | F-ATPase protein 6 (*ATP6*) | 7.93 | 8.41 | 0.67 |
| c122818_g1 | cytochrome P450 17A1 (*CYP17A1*) | 0.81 | 1.63 | 0.82 |
| **DEGs related to lipid metabolic process** | | | | |
| c183562_g1 | apolipoprotein E precursor (*APoE*) | 4.59 | 4.18 | −0.40 |
| c108401_g1 | coenzyme Q-binding protein (*CoQ10*) | 2.51 | 3.44 | 0.93 |
| c106996_g1 | high-density lipoprotein particle (*SAA*) | 1.24 | 4.13 | 2.89 |
| c101752_g1 | alcohol dehydrogenase 4 (*ADH4*) | −3.8 | – | 3.81 |
| c5275_g1 | fatty acid desaturase 6 (*FADS6*) | −3.38 | −1.54 | 1.85 |
| **DEGs related to carbohydrate metabolic process** | | | | |
| c184502_g1 | hexokinase-1 (*HK*) | 4.49 | 5.22 | 0.73 |
| c141564_g1 | lens fiber major intrinsic protein (*MIP*) | 3.37 | 3.75 | – |
| c121000_g1 | glucokinase (*GCK*) | −0.22 | −1.94 | −1.85 |
| c198426_g1 | tagatose 1,6-diphosphate aldolase (*LacD*) | −5.38 | −5.12 | – |
| c139209_g1 | L-gulono-gamma-lactone oxidase (*GLO*) | −5.38 | −5.12 | – |
| c44501_g1 | ribulose bisphosphate carboxylase small chain (*RBCs*) | −4.38 | −4.12 | – |
| **DEGs related to amino acid metabolic process** | | | | |
| c198734_g1 | ornithine decarboxylase antizyme 3 (*OAZ3*) | 6.28 | 7.00 | 0.72 |
| c184545_g1 | glutamine synthetase (*GLUL*) | 4.03 | 4.3 | 0.26 |
| c197142_g1 | asparaginase-like protein 1b (*ASRGL*) | 4.49 | 4.96 | 0.47 |
| c117856_g3 | L-amino-acid oxidase-like (*LAAO*) | 1.62 | −2.06 | −3.68 |
| c169209_g1 | sodium-dependent neutral amino acid transporter B (*SLC6A15s*) | 3.59 | – | −1.54 |
| c103361_g1 | amino acid permease (*SLC7A9*) | 1.78 | 5.72 | 3.93 |
| c108456_g1 | tyrosine aminotransferase (*TAT*) | −1.72 | −1.41 | 0.31 |
| c99414_g1 | argininosuccinate synthase (*ASS1*) | −0.69 | −1.1 | −0.41 |
| **DEGs related to ion transport** | | | | |
| c182997_g1 | potassium voltage-gated channel subfamily H member 5 (*KCNH5*) | 3.5 | – | −3.5 |
| c121057_g1 | serine/threonine-protein kinase 32 (*STK32*) | 1.95 | 1.80 | – |
| c121806_g3 | salt-inducible kinase 1 (*SIK1*) | 1.23 | 2.20 | 0.97 |
| c123793_g3 | salt-inducible kinase 2 (*SIK2*) | – | 0.27 | 0.40 |
| c114797_g4 | adiponectin (*ACDC*) | 0.60 | 2.12 | 1.52 |
| c119528_g1 | insulin receptor-related protein (*INSRR*) | 0.30 | −1.70 | −2.00 |
| c95283_g1 | serine/threonine-protein kinase 33 (*STK33*) | −1.5 | −0.65 | 0.85 |
| c58533_g1 | sodium channel subunit beta-1 (*SCN1B*) | −5.31 | −5.31 | – |
| c94169_g1 | natriuretic peptides A-like (*NPPA*) | −5.72 | −5.72 | – |
| c95302_g1 | solute carrier family 26 member 9 (*SLC26A9*) | −3.55 | −0.44 | 3.11 |

**Notes.**
The values above zero show up-regulation of gene expression, while the values below zero show down-regulation. "–" means that the level of gene expression is so low that it could not be detected.

aminotransferase, *TAT* and argininosuccinate synthase, *ASS1*) showed down-regulation in the comparison of 0 vs. 5 psu, while four DEGs (*OAZ3*, *GLUL*, *ASRGL*, *SLC7A9*) showed up-regulation and three DEGs (*LAAO*, *TAT*, *ASS1*) showed down-regulation in the comparison of 0 vs. 15 psu.

### Verification of gene expression by SYBR Green qRT-PCR

Some genes related to ion transport, energy production and conversion and macromolecule metabolic process mentioned above were selected for qRT-PCR analysis in order to validate the differentially expressed genes that were identified by RNA-Seq and gain detailed quantitative information on their differing expression patterns. As shown in Table 7, the mRNA expression levels of *COX3*, *ATP6* and *CYP17A1* related to energy production and conversion in the 5 psu group were almost 3.6-, 2.2- and 1.5-fold of the control respectively, while those in the 15 psu group were almost 2.4-, 3.4- and 2.1-fold of the control respectively. In relation to ion transport, the mRNA expression levels of *STK32* and *SIK1* in the 5 psu group were almost 3.6- and 2.1-fold of the control respectively, while those in the15 psu group were almost 3.0- and 3.5-fold of the control respectively. The mRNA expression levels of *INSRR* increased 2-fold in the 5 psu group and decreased 2.6-fold in the 15 psu group compared to the control, however, that of *STK33* decreased 2.2- and 1.5-fold in the group of 5 psu and 15 psu compared to the control. As for macromolecular metabolism, the mRNA expression levels of *APoE*, *CoQ10*, *SAA*, *HK*, *GLUL*, and *ASRGL* increased with ambient salinity increased. Especially for the genes of *CoQ10* and *SAA*, the mRNA expression levels in the 15 psu group were almost 8.4- and 15.3-fold of the control respectively. However, the mRNA expression levels of *GCK* and *ASS1* decreased with ambient salinity increased, and those in the group of 5 psu were almost 1.5- and 2.0-fold of the control, and those in the group of 15 psu were 3.7- and 9.7-fold of the control. The mRNA expression levels of *FADS6* and *TAT* in the 5 psu group were lowest among the three groups, and decreased almost 2.3- and 3.6-fold compared to the control.

The qRT-PCR results were significantly correlated with the RNA-seq results with correlation coefficients of 0.744 in 0 vs. 5 psu, 0.862 in 0 vs. 15 psu group and 0.748 in 5 vs. 15 psu group ($p < 0.05$) (Fig. S1). This provides strong evidence that RNA-Seq data can be a reliable indicator of the expression patterns of the hundreds of genes identified as differentially expressed in the current study.

## DISCUSSION

Osmoregulation in some aquatic animals can be a complex process because individuals must deal with fluctuating salinity levels in their natural habitats, often on a daily or seasonal basis. As a normally freshwater species, the red-eared slider *T. s. elegans*, does not possess salt glands, and requires osmoregulation to survive when entering environments of higher salinity (e.g., brackish or estuarine water). The physiological research has shown that *T. s. elegans* can increase blood osmotic pressure by balancing the entry of NaCl with the decreased secretion of aldosterone, and accumulating urea and free amino acids in blood (*Hong et al., 2014*). In our study, many DEGs were classified into some dominant categories including the macromolecule metabolic process, ion transport and ion transmembrane

**Table 7  The expression levels of some genes in relation to osmotic adjustment in the liver of *T. s. elegans* by qRT-PCR.**

| Unigene ID | Description | Control | 5 psu | 15 psu |
|---|---|---|---|---|
| **Genes related to energy production and conversion** | | | | |
| c205807_g1 | cytochrome c oxidase subunit III (*COX3*) | 1.01 ± 0.06[c] | 3.57 ± 0.27[a] | 2.40 ± 0.29[b] |
| c168776_g1 | F-ATPase protein 6 (*ATP6*) | 1.09 ± 0.10[c] | 2.42 ± 0.09[b] | 3.67 ± 0.21[a] |
| c122818_g1 | cytochrome P450 17A1 (*CYP17A1*) | 2.13 ± 0.11[c] | 3.15 ± 0.22[b] | 4.46 ± 0.38[a] |
| **Genes related to lipid metabolic process** | | | | |
| c183562_g1 | apolipoprotein E precursor (*APoE*) | 0.86 ± 0.06[c] | 2.77 ± 0.12[a] | 1.80 ± 0.13[b] |
| c108401_g1 | coenzyme Q-binding protein (*CoQ10*) | 0.64 ± 0.08[c] | 2.66 ± 0.17[b] | 5.39 ± 0.23[a] |
| c106996_g1 | high-density lipoprotein particle (*SAA*) | 0.48 ± 0.06[c] | 1.18 ± 0.08[b] | 7.32 ± 0.27[a] |
| c5275_g1 | fatty acid desaturase 6 (*FADS6*) | 2.23 ± 0.27[a] | 0.97 ± 0.23[b] | 1.37 ± 0.13[b] |
| **Genes related to carbohydrate metabolic process** | | | | |
| c184502_g1 | hexokinase-1 (*HK*) | 1.37 ± 0.09[c] | 2.35 ± 0.21[b] | 3.27 ± 0.22[a] |
| c121000_g1 | glucokinase (*GCK*) | 1.79 ± 0.07[a] | 1.19 ± 0.22[b] | 0.48 ± 0.05[c] |
| **Genes related to amino acid metabolic process** | | | | |
| c184545_g1 | glutamine synthetase (*GLUL*) | 1.82 ± 0.14[c] | 5.10 ± 0.23[b] | 6.00 ± 0.19[a] |
| c197142_g1 | asparaginase-like protein 1b (*ASRGL*) | 1.01 ± 0.05[b] | 1.64 ± 0.17[a] | 1.94 ± 0.31[a] |
| c108456_g1 | tyrosine aminotransferase (*TAT*) | 3.37 ± 0.17[a] | 0.93 ± 0.03[c] | 1.38 ± 0.06[b] |
| c99414_g1 | argininosuccinate synthase (*ASS1*) | 5.16 ± 0.29[a] | 2.59 ± 0.20[b] | 0.53 ± 0.07[c] |
| **Genes related to ion transport** | | | | |
| c121806_g3 | salt-inducible kinase 1 (*SIK1*) | 1.19 ± 0.13[c] | 2.46 ± 0.31[b] | 4.16 ± 0.15[a] |
| c123793_g3 | salt-inducible kinase 2 (*SIK2*) | 1.06 ± 0.15[a] | 0.85 ± 0.09[a] | 1.13 ± 0.10[a] |
| c119528_g1 | insulin receptor-related protein (*INSRR*) | 2.04 ± 0.18[b] | 4.13 ± 0.26[a] | 0.78 ± 0.13[c] |
| c121057_g1 | serine/threonine-protein kinase 32 (*STK32*) | 1.48 ± 0.16[c] | 5.39 ± 0.24[a] | 4.49 ± 0.33[b] |
| c95283_g1 | serine/threonine-protein kinase 33 (*STK33*) | 3.93 ± 0.37[a] | 1.79 ± 0.16[b] | 2.57 ± 0.23[b] |

**Notes.**
Different lowercase letters represent significance among different groups ($p < 0.05$).

transport in biological process, which provide a strong evidence for the physiological mechanism.

Under salinity stress, loss of water from the cells can cause cells shrink and potentially die. In this sense, ion regulation is important for a cell to balance osmotic change. *KCNH5* (http://www.ncbi.nlm.nih.gov/gene/27133) and *SCN1B* (*Qin et al., 2003*) are involved in potassium/sodium voltage-gated ion channels and fluid balance that controls arterial blood pressure by altering blood electrolyte composition and/or volume. Natriuretic peptide A (*NPPA*) is well known to regulate body fluid levels and electrolytic homeostasis and has natriuretic, diuretic, and vasodilatory actions (*Espiner et al., 2014*). *NPPA* is highly expressed and associated with $H_2O/Na^+$ absorption and protein Ser/Thr phosphatases (*Espiner et al., 2014*). *SIK* acts to modulate adrenocortical function particularly in response to high plasma $Na^+$, $K^+$, ACTH, or stress (*Wang et al., 1999*). *SIK1* also has a role in steroidogenesis whereas *SIK2* is implicated in gluconeogenesis regulation in liver; both belong to the AMPK (AMP-activated kinase) subfamily of serine/threonine kinases (*Berggreen et al., 2012*). The AMPK is a crucial regulator of cellular energy levels (*Hardie & Ashford, 2014*) and under stress conditions that deplete ATP, AMPK action promotes ATP-producing catabolic pathways while inhibiting ATP-consuming anabolic functions (*Rider*

*et al., 2009*; *Rider et al., 2006*). Transcripts of adiponectin (*ACDC*) were also enriched in liver under both salinity stresses and this hormone participates in the pathway of fatty acid oxidation by regulating AMPK (*Chong et al., 2013*). So, in our study, five genes (*ACDC, INSRR, STK32, SIK1, KCNH5*) were up-regulated in the comparison of 0 vs. 5 psu, and three genes (*ACDC, STK32, SIK2*) were up-regulated in the comparison of 0 vs. 15 psu, which suggested that *T. s. elegans* can adapt itself into saline environment by increasing the expression levels of genes related to ion regulation.

Genes associated with transporting molecules related to metabolic processes were also modulated in association with an up-regulation of genes involved in ATP energy production. As salinity level increases, acclimation of *T. s. elegans* to elevated salinity conditions requires investment of energy to maintain their homeostatic balance (*Hong et al., 2014*). In our study, the DEGs related to energy production and conversion including *ATP6, COX1, COXIII, CYTb,* and *CYP17A1* increased with ambient salinity increased, which suggested that a requirement for more energy by *T. s. elegans* during raised salinity conditions. The result is associated with the ATP needed for the synthesis and operation of transport-related proteins that drive ion- and osmoregulatory processes (*Lee et al., 2003*). It is also documented that changes in components related to the glycolysis, fatty acid metabolism, and ATP production are often associated with raised salinity conditions in freshwater fish (*Lavado, Aparicio-Fabre & Schlenk, 2014*; *Tine et al., 2008*). Our previous study has shown that *T. s. elegans* increased serum glucose and triglyceride levels when subjected to salinity stress (*Shu et al., 2012*). This study also indicated that salinity stress influences glycolysis/gluconeogenesis and fatty acid metabolism-related genes. The mRNA expressions of *HK* and *MIP* were increased, which suggested that utilization of glycogen might be increased and more glucose might be used for energy expenditure under salinity exposure. Moreover, the gene expressions of *APoE, CoQ10,* and *SAA* were up-regulated while *ADH4* and *FADS6* were down-regulated, which suggested that the lipolysis increased and lipogenesis decreased to produce more energy in response to salinity stress.

The ureogenesis is a strategy for diamondback terrapins (*Malaclemys terrapin* ) that are known to inhabit brackish water, and for the desert tortoise, *Gopherus agassizii* under dehydration stress (*Dantzler & Schmidt-Nielsen, 1966*). The red-eared slider *T. s. elegans*, can also increase urea content in plasma and tissues by synthesizing organic osmolytes (e.g., free amino acids and urea) in order to provide a colligative defense against water loss (*Hong et al., 2014*). Some DEGs associated with cellular amino acid metabolism including *GLUL, ASRGL, LAAO, SLC6A15s* and *SLC7A9* were up-regulated in the comparison of 0 vs. 5 psu, which indicated that amino acid metabolism and transport were strengthened by salinity stress. The up-regulated expression of *GLUL* showed the increase of glutamine synthetase. Glutamine is the most prevalent amino acid in body fluids and muscle, is mainly transported by a $Na^+$-dependent neutral amino acid system, and its turnover rate exceeds those of other amino acids (*Zander et al., 2015*). Glutamine is synthesized from the ATP-dependent conjugation of ammonia to glutamate, and is a well-known defense against ammonia (*Cooper & Plum, 1987*; *Essexfraser et al., 2005*). When salinity returns to normal, glutamine pools can be utilized as precursors for a variety of important cell molecules (e.g., purines, pyrimidines, mucopolysaccharides) or, in the presence of glutaminase, glutamine

can be deaminated for direct excretion of ammonia in the kidney or used for urea synthesis in liver before excretion. The accumulation of urea in response to high salinity can be due to both urea retention and elevated rates of urea synthesis via the ornithine-urea cycle, uricolysis of uric acid, or hydrolysis of arginine (*Dépêche & Schoffeniels, 1975*; *Gordon & Tucker, 1965*). *OAZ3*, involved in the ornithine-urea cycle, showed up-regulation under salinity exposure in this study, which suggested that ureogenesis may also be activated in response to hyperosmotic conditions in this species. Our result correlated well with studies in crab-eating frog (*Fejervarya cancrivora*), where its resistance to hyperosmotic environmental conditions is generally linked to the accumulation of urea in plasma and tissues, urea resulting from an up-regulation of the hepatic urea synthesis machinery (*Gordon & Tucker, 1965*). Also, the striped catfish (*Pangasianodon hypophthalmus*) can counteract osmotic imbalance by triggering a regulatory volume increase, an internal process that initiates a net gain in osmolytes and/or water, increasing cell volume that re-establishes normal values and prevents further cell shrinkage (*Nguyen et al., 2016*).

The current study was designed not only to generate a catalogue of differentially expressed genes involved with salinity exposure, but also to allow the data to be integrated to identify the relationships on the adaptive response. Based on gene ontology information and from data in published studies, functional categories of differentially expressed genes were identified. Therefore, the potential interactions of the differentially expressed genes that responded to high salinity in *T. s. elegans* are shown in Fig. 2. The pathways and processes that are targeted provide us with numerous candidate genes for future investigations about the molecular mechanisms that underlie high salinity tolerance.

## CONCLUSIONS

In this study, we report the first transcriptome analysis of *T. s. elegans* under salinity stress. When *T. s. elegans* was subjected to salinity exposure, 3,393 unigenes in the liver were identified as DEGs, which were classified into some dominant categories including macromolecule metabolic process, ion transport, ion transmembrane transport in biological process. Also, a list of candidate DEGs potentially involved in salinity acclimation in *T. s. elegans* were identified into three kinds such as ion transport, energy production and conversion, and macromolecule metabolic process including lipid, protein and carbonhydrate.

The genes related to macromolecule metabolic process (*OAZ3*, *GLUL*, *ASRGL*, *LAAO*, *SLC6A15s*, *SLC7A9*, *APoE*, *CoQ10*, *SAA*, *HK*, and *MIP*), ion transport (*KCNH5*, *STK32*, *SIK1*, and *ACDC*) and ATP synthesis (*ATP6*, *COX1*, *COX3*, *CYTb*, and *CYP17A1*) were up-regulated by salinity stress, which indicated that *T. s. elegans* could adapt itself into salinity stress by balancing the entry of NaCl and accumulating urea and free amino acids in blood in response to osmotic pressure with higher ATP energy production. However, some genes related to macromolecule metabolic process (*ADH4*, *FADS6*, *LacD*, *GLO*, *RBCs*, *TAT*, and *ASS1*) and ion transport (*SCN1B*, *NPPA*, *SLC26A9*, and *STK33*) were down-regulated by salinity stress. Finally, we combined the data on functional salinity tolerance genes into a hypothetical schematic model that describe potential relationships

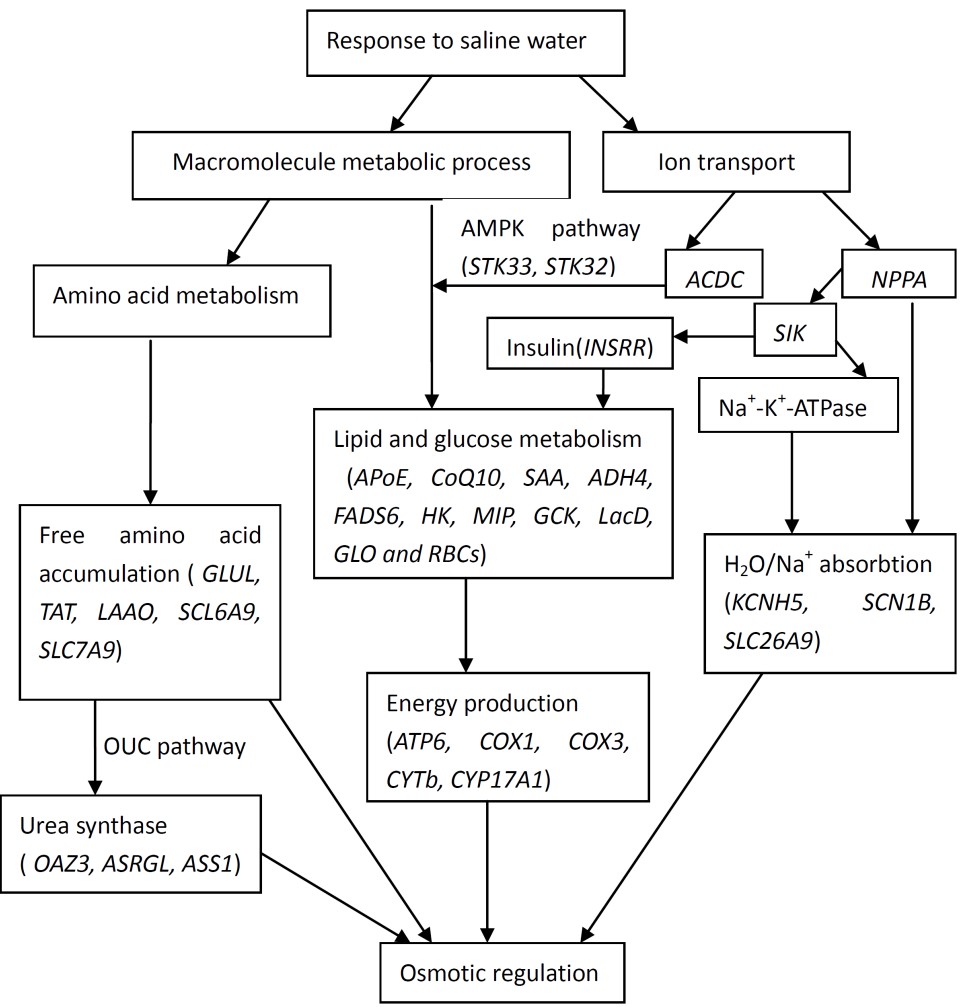

**Figure 2** **Interactions of positively selected genes and differentially expressed genes involved in the adaptation of *T. s. elegans* to high salinity.** (1) *NPPA* plays roles in the regulation of body fluid levels and electrolytic homeostasis pathway, while *ACDC* and *SIK* in the pathway of lipid (glucose) metabolism by the regulation of *AMPK* (*STK32*, *STK33*); (2) *GLUL*, *TAT*, *LAAO*, *SCL6A9*, *SLC7A9* and *OAZ3*, *ASRGL*, *ASS1* play roles in the accumulation of free amino acid and urea, while *KCNH5*, *SCN1B*, *SLC26A9* mainly in the process of $H_2O/Na^+$ absorption; (3) *ATP6*, *COX1, COX3, CYTb, CYP17A1* are associated with energy production and mediated by lipid and glucose metabolism (*APoE, CoQ10, SAA, ADH4, FADS6, HK, MIP, GCK, LacD, GLO and RBCs*).

and interactions among target genes to explain the molecular pathways related to salinity responses in *T. s. elegans*.

## ACKNOWLEDGEMENTS

We are grateful to Dr. Heidy Kikillus from Victoria University of Wellington, New Zealand for revising a previous version of this manuscript.

### Funding

This work was supported by the National Natural Science Foundation of China (grant numbers 31760116, 31360642, 31372228). The funders had no role in study design, data collection and analysis, decision to publish, or preparation of the manuscript.

### Grant Disclosures

The following grant information was disclosed by the authors:
National Natural Science Foundation of China: 31760116, 31360642, 31372228.

### Competing Interests

Kenneth B. Storey is an Academic Editor for PeerJ.

### Author Contributions

- Meiling Hong conceived and designed the experiments, performed the experiments, analyzed the data, contributed reagents/materials/analysis tools, prepared figures and/or tables, authored or reviewed drafts of the paper, approved the final draft.
- Aiping Jiang conceived and designed the experiments, performed the experiments, analyzed the data, contributed reagents/materials/analysis tools, approved the final draft.
- Na Li, Weihao Li and Haitao Shi performed the experiments, approved the final draft.
- Kenneth B. Storey conceived and designed the experiments, analyzed the data, authored or reviewed drafts of the paper, approved the final draft.
- Li Ding conceived and designed the experiments, analyzed the data, contributed reagents/materials/analysis tools, prepared figures and/or tables, authored or reviewed drafts of the paper, approved the final draft.

### Animal Ethics

The following information was supplied relating to ethical approvals (i.e., approving body and any reference numbers):

Aninal Research Ethics Committee of Hainan Provincial Education Center for Ecology and Environment, Hainan Normal University provided full approval for this research (HNECEE-2014-004).

### Data Availability

The raw data can be found under accession number GSE117354.

### Supplemental Information

Supplemental information for this article can be found online at http://dx.doi.org/10.7717/peerj.6538#supplemental-information.

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
