# Peer review of "Comparative analysis of the liver transcriptome in the red-eared slider Trachemys scripta elegans under chronic salinity stress"

_PeerJ, doi:10.7717/peerj.6538_

## Round 0.1 · original submission · Major Revisions

Please address all the critical issues raised by the reviewers and amend manuscript accordingly.

·

Basic reporting

This study uses new laboratory methods (radseq, qpcr) and analyses (GO analysis, KEGG analysis) to try to understand the molecular pathway of salinity adaptation in a highly invasive species Trachemy scripta elegans. The paper is well written, but would benefit from some clarification in some areas. I have attached a pdf file with edits and suggestions. The literature references, as well as background provided are sufficient. I highlight a key point below:
- for the keywords, try to avoid using words that are in the title, as the title is also used as keywords in online searches. I have added some keywords that I think are appropriate.

Experimental design

This research is original and within the aims and scope of the journal. The research question is well defined and relevant to studying the cellular reaction to salinity stress, as well as invasive species. We know that T. s. elegans can live in saline habitats, and this study begins to answer the “how” question. One clarification that is needed on the experimental design:
- For methods, it is a bit confusing how many individuals were used in the study. 9 total or 18 total? This is a key point that needs to be clarified

Validity of the findings

The discussion, overall its good, but could improve. The biggest thing I thought was missing was the "why is this important?" question. You did a good job describing the results, but I think you could take it a step further. First is the identification of the key genes in TSE. After listing these genes, what next? One thing you can emphasize more are the comparisions. 0 vs 5, 0 vs 15. The genes that are shared between these comparisons are interesting--these shared pathways indicate some importance to salinity stress. Also, are there any big changes from 5 vs 15? you touched on this with the discussion of hormesis, but could be elaborated. After this, you can compare your study to the study of other studies of salinity stress. You did a good job of this, but could be elaborated a bit more. Any other studies of salinity stress that sequenced transcriptomes? Anything else you can add when comparing to the diamondback terrapin? This is a good comparison since it is also a turtle. Lastly, what are the implications of your work to understanding the invasive nature of this species? Does this help us with controlling the species? Does it indicate anything we should be careful of? This would be a nice thing to add towards the end, increase the usefulness of your study.

Reviewer 2 ·

Basic reporting

BASIC REPORTING
1) Clear, unambiguous, professional English language used throughout
This manuscript is exceptional in its use of well-crafted English sentences. It was easy to read in that respect.
2) Intro & background

The authors lay the foundation for this report as follows. The fresh water red eared slider turtle is an invasive species whose success is due, at least in part, to its ability to invade saline environments. They state there are really two ways to deal with changing salinity (Line 63) “tolerance of elevated inorganic ion concentrations (mainly sodium and chloride) in plasma (Gordon & Tucker 1965), and accumulation of organic osmolytes (e.g. urea) to counteract cell-volume changes”. First, I would say that, as written, these are not two things, but one: tolerance could be achieved by accumulation of organic osmolytes to keep cell volumes the same.

But the two strategies are actually 1) keeping plasma osmolarity constant and 2) altering cell osmolarity to match plasma osmolarity to maintain cell volume. While it is possible that this turtle allows plasma chloride to change and responds by synthesizing organic osmolytes, the authors have previously reported increase in urine osmolarity and drop in aldosterone, consistent with the hypothesis that ability to tolerate higher salt in the environment is also achieved by homeostatic responses that act to maintain serum osmolarity at a constant level. This is what sea turtles do, and this strikes me as much more germane to this species than what crabs do. Routes of excretion should be discussed.

Commitment to exploring this particular strategy can then be put in context of their 2014 paper which also suggests that tolerance strategies are involved. The authors need to establish that they were looking for

3) Structure conforms to PeerJ standards, discipline norm, or improved for clarity.

The structure was very good.

4) Figures are relevant, high quality, well labelled & described.

The paper was very well written, and I got to the end of the manuscript without noticing the figures. Only one figure – figure 4 – is mentioned in the text of the manuscript. If figures are not mentioned in the text, they obviously add nothing to the manuscript and should be removed.

Figure 1 does show that differential gene expression occurs. The colors used in Figure 1 are mostly a distraction, as they map directly to the x and y scales.

I recommend a Venn diagram showing overlap between significant responses at the two salinities.

Figure 2 seems to show that genes differentially expressed genes map to many categories that have no specific connection to this experiment or the hypothesis that this turtle responds to salinity stress by increasing synthesis of organic osmolytes.

Figure 3 needs to be discussed. Why did the authors pick genes at random? If this paper is about synthesis of osmolytes in the liver in response to salt stress, the authors should do qPCR on the genes they believe are induced or repressed that explain the tolerant phenotype, rather than showing that RNA-seq works, whichis well accepted.

Figure 4 is highly speculative and highly flawed, in my view. On what basis do the authors conclude that “hormonal system” (whatever that is) drives the osmotic stress response in this turtle leading to “plasma osmotic pressure”? This appears to be a reference to their previous work showing changes in aldosterone. To make a figure like this work, they will need to create a pathway of related genes that are significantly differentially expressed, that are known to be co-regulated. For example, a transcription factor might be known to induce the expression of some gene that is part of a signaling pathway that eventually leads to increase synthesis of a particular osmolyte. The figure needs to show the direction of observed regulation and arrows showing how that gene is predicted to influence the expression of other genes in their pathway. Since this a gene expression study, there is no room here for vague concepts like “energizer”. All genes have been measured. Show how the genes work together to achieve the phenotype.



5) Raw data supplied (see PeerJ policy).

I was not able to access any raw data.

Experimental design

1) Original primary research within Scope of the journal.

Yes.

2) Research question well defined, relevant & meaningful. It is stated how the research fills an identified knowledge gap.

The authors set out to explore the molecular basis of salinity tolerance in this turtle and found some genes that are differentially expressed that support a pre-existing hypothesis, namely, that this organism tolerates high salinity because it synthesizes osmolytes that protect it from cell volume changes. Unfortunately, it’s really just an exploration of various changes that occur, most of which have nothing at all to do with their hypothesis. I see that as more a strength than a weakness, but I would like the authors to clarify that they saw many exciting differences in the liver, suggesting a huge response, and that some of the gene expression changes are consistent with their previous hypothesis, though many are not directly related in an obvious way.

What are the key genes that responded in a way consistent with their hypothesis? I’d like a graph showing how they responded (e,g. fold change) comparing 0, 5 and 15 psu gene expression, with error bars, or the raw data points.

3) Rigorous investigation performed to a high technical & ethical standard.

Ethical standards met. Given that they performed an experiment with two levels of osmotic challenge, why comparisons between them not addressed?

4) Methods described with sufficient detail

I would like clarification on the following technical issues:

- Why was the liver chosen?
- Why was beta actin chosen as a reference gene (was it differentially expressed?)
- Why was a fold change of 2 and an FDR of 0.05 chosen?
- How was significance established in GO term and KEGG pathway enrichment?
- Were these GO term and KEGG pathway enrichment p values corrected for multiple hypotheses?
- How many reads were in each of the samples? Were the library sizes the same across all samples?
- What does it mean for the transcriptomic assembly to result in 200,000 unigenes? How does this compare to your expectations?

Validity of the findings

This paper begins with a core hypothesis that needs to be stated. I believe that hypothesis is that gene expression patterns in the liver would demonstrate a response to salinity consistent with tolerance achieved by increased osmolyte synthesis. I do not believe the authors have provided persuasive evidence of this, but I think a careful cherry-picking of the data can make this a plausible argument.

Alternatively, the authors can simply characterize this an early exploration of this response. This is useful work to this field and it’s very hard to use gene expression alone to establish molecular mechanisms. In my opinion, recognizing the limitations of their design would make their new hypotheses based on gene expression more compelling, not less. But I’d like more detail about what genes they believe are driving this, based only on gene expression differences they measured here.

Finally, I’d like the authors to speculate about how their results inform the invasive species problem they lay out in their introduction. They know more (we hope) now what do we do?

Additional comments

Good luck. I think this is good work and should get into print.

---

## Round 0.2 · Major Revisions

As you can see, both reviewers feel that additional revision is required. Therefore, please address their critical issues and amend your manuscript accordingly.

·

Basic reporting

good

Experimental design

good

Validity of the findings

good

Additional comments

I am happy with the modifications made by the authors. One thing to be careful of is the formatting in the paper. There needs to be consistency.

For example, when making a list, do you include a comma before the "and"? It seemed that you did this more often than not, so I tried to add the comma whenever missing, but please read through the manuscript to make sure it is consistent throughout the paper.

Also, the formatting of your numbers is inconsistent. For example, are numbers written
1) 1,234
2) 1 234
3) 1234
This may be something that the editor can decide for you based on journal formatting. Make sure it is consistent throughout.

I have re-read and tried to clean up the language. I attached the manuscript with track changes.

Reviewer 2 ·

Basic reporting

Are Raw data supplied ?

Data are now available in GEO. Although the text in the manuscript indicates that nine turtles were used, suggesting n=3 per group, only 3 samples (total) are stored in GEO. This simple fact explains much of what needs to be changed in the manuscript. Although it is hard to do statistics with 3 replicates per group, few would agree that one can do statistics with n=1 per group.

 It is vital that the manuscript clearly state that there were no replicate measurements.
In addition, though the data cached in GEO does include count tables, which is helpful in principle. However, it is not possible to map identifiers in the count tables to anything mentioned in the manuscript. Therefore, the count tables are not useful. Information should be supplied to allow other scientists to use these data without having to perform a new de novo alignment.

Are Figures relevant, high quality, well labelled & described?

The figures collectively illustrate the following story. 1) Similar differences were observed between 0-5, 0-15, and 5-15 ppt. 2) About 30 KEGG pathways had differentially expressed genes in them. 3) Fold changes based on qPCR and RNA-seq are strongly correlated. 4) It is possible to construct a diagram using carefully selected genes that supports a (supposedly causal) relationship starting with “response to salt water” and ending with “plasma osmotic pressure.”

Figure 1 is blurry. It obviously has no relevance, because it is not mentioned in the text. However, it does raise several important questions. How were the FDR corrected p-values garishly highlighted in this figure actually calculated in an n of 1 experiment? Given the very large number of “highly significant” genes identified in each comparison, were the same genes significant in each comparison? Were genes that were significant in multiple comparisons differentially expressed in the same direction in different conditions relative to control? Were the same or different GO groups and KEGG pathways enriched at 5 and 15 ppt? Is a dose response evident? Is the 15ppt challenge suggest that the turtles are more stressed than at 5 ppt?

Figure 2 is meaningless without indicating the fraction of genes induced and repressed. If 100% of genes associated with a particular pathway are induced in 5 ppt compared to 0 ppt, one can infer that the pathway as a whole is activated at this concentration. On the other hand, if half of the genes in the pathway are up and half are down, it is hard to know how the organism is responding. As mentioned above, this figure should compare and contrast 0-5, 0-15, and 5-15 ppt.
 This figure needs to be broken into three panels, one for each contrast in Figure 1, and the number of induced and repressed genes indicated.

Figure 3 Supports the conclusion that RNA-seq works, which would have been relevant many years ago, but is no longer in doubt. Contemporary RNA-seq papers do not waste space demonstrating that RNA-seq and qPCR yield similar results. The point of RNA-seq verification is to demonstrate that the genes central to the interpretation chosen by the authors (e.g. those in Figure 4) that were observed in a high dimension experiment (RNA-seq) can be replicated in a low dimension experiment. Randomly selected genes are irrelevant.

Here is a recent paper illustrating the use of qPCR validation and a mechanism and how Figure 2 should show fraction of induced and repressed genes : https://doi.org/10.1016/j.jhazmat.2018.11.101

 Figure 3 must be replaced by a new figure showing the genes selected in Figure 4. Each replicate turtle must be used (n = 3) and statistics performed.

Data is robust, statistically sound, & controlled.

The manuscript suggests that 9 turtles were used, in three experimental groups (0, 5, 15) and that statistics were performed using edgeR, yielding roughly 3,000 genes achieving an FDR < 0.05 and an absolute log2 fold change of 1 or more. In fact, only 1 replicate has been used. Although a kind of statistic can be generated by edgeR, it is specifically not recommended by the authors. The following is from the edgeR manual:

2.11 What to do if you have no replicates
edgeR is primarily intended for use with data including biological replication. Nevertheless, RNA- Seq and ChIP-Seq are still expensive technologies, so it sometimes happens that only one library can be created for each treatment condition. In these cases there are no replicate libraries from which to estimate biological variability. In this situation, the data analyst is faced with the following choices, none of which are ideal. We do not recommend any of these choices as a satisfactory alternative for biological replication. Rather, they are the best that can be done at the analysis stage, and options 2–4 may be better than assuming that biological variability is absent.
1. Be satisfied with a descriptive analysis, that might include an MDS plot and an analysis of fold changes. Do not attempt a significance analysis. This may be the best advice.


 If the authors are using one of the mechanisms defined in the user guide (but not recommended) to overcome lack of replication, they must be completely candid about this or characterize the analysis as descriptive.

Experimental design

No Comment

Validity of the findings

Impact and Validity of Findings

The findings presented in this paper have not been adequately supported by the data, statistics and figures. Aside from the issues raised above, the manuscript presents itself as exploring how an invasive species invades by improved salinity acclimation.

“The [sic] physiological research has shown that T. s. elegans could increase blood
osmotic pressure by balancing the entry of NaCl with the decreased secretion of aldosterone, and accumulating urea and free amino acids in blood (Hong et al. 2014). Results of GO enrichment analysis in the study showed that ion binding, ion transmembrance transport, hydrogen ion transmembrane transporter activity, hydrogen ion transmembrane transport, hormone activity, monovalent inorganic cation transport, anion binding, and ion transport were significantly differentiated, which provide a strong evidence for the physiological mechanism. “

Which physiological mechanism? In the foregoing passage it sounds as though ion transport in the liver might be expected to play a role in salinity acclimation. How are they proposing this might work?

If the idea is that acclimation is accompanied by changes in the urea cycle, mentioned more than once in the manuscript, it follows that many genes associated with this pathway would be induced at high salinities. As this was not reported, I assume it was not observed.

Annotated reviews are not available for download in order to protect the identity of reviewers who chose to remain anonymous.

---

## Round 0.3 · accepted · Accept

In my view, the authors adequately addressed all the critical issues raised by the reviewers and revised their manuscript accordingly. Therefore, the amended version can be published in its current form.

# Reviewer 2 ·

Basic reporting

No further comments.

Experimental design

No further comments.

Validity of the findings

No further comments.

Additional comments

Thanks for your hard work.